# Comparison of Chiggers (Acari: Trombiculidae, Leeuwenhoekiidae) on Two Sibling Mouse Species, *Apodemus draco* and *A. ilex* (Rodentia: Muridae), in Southwest China

**DOI:** 10.3390/ani13091480

**Published:** 2023-04-27

**Authors:** Yu Guo, Xian-Guo Guo, Wen-Yu Song, Yan Lv, Peng-Wu Yin, Dao-Chao Jin

**Affiliations:** 1Institute of Pathogens and Vectors, Yunnan Provincial Key Laboratory for Zoonosis Control and Prevention, Dali University, Dali 671000, China; 2Institute of Entomology, Guizhou University, Guiyang 550025, China

**Keywords:** chigger mites, ectoparasite, South China field mouse, Lantsang field mouse, rodent, southwest China

## Abstract

**Simple Summary:**

Chigger mites (Acari) are common ectoparasites and the exclusive vector of scrub typhus. South China field mouse (*Apodemus draco*) and Lantsang field mouse (*A. ilex*) are two sibling rodent species. The investigation in southwest China (2001–2015) showed that chigger infestations on two mouse species were quite different, including different species composition, overall infestation, community parameters and dominant chigger species. There were 36 chigger species found on *A. draco* and 11 on *A. ilex*, and the overall mean intensity of chiggers on *A. draco* (*MI* = 4.26) was higher than that on *A. ilex* (*MI* = 3.91, *p* < 0.05). Dominant chiggers were unevenly distributed among different individuals of mouse hosts, and chigger infestation showed sex bias on different mouse sexes. The species abundance of the chigger community on *A. draco* was revealed as a log-normal distribution pattern.

**Abstract:**

Chigger mites (Acari) are common ectoparasites on rodents, and they are the exclusive vector of scrub typhus. South China field mouse (*Apodemus draco*) and Lantsang field mouse (*A. ilex*) are two sibling rodent species. Based on field investigations in southwest China (2001–2015), this paper compared the infestation of these two mouse species with chiggers. Of 42 chigger species identified from two mouse species, 36 were found on *A. draco*, 11 on *A. ilex* and 5 common species on both mice. Jaccard similarity index (*J* = 0.12, *J* < 0.25) showed a very different species composition of chiggers on two mouse species, and some parameters of the chigger community were also different. The overall mean intensity of chiggers on *A. draco* (*MI* = 4.26) was higher than that on *A. ilex* (*MI* = 3.91, *p* < 0.05). The dominant chigger species on *A. draco* were *Trombiculindus yunnanus*, *Leptotrombidium scutellare* (a major vector species in China) and *L. sinicum* with a total constituent ratio *C_r_* = 42.9% (106/247). *Leptorombidium sinicum* and *L. scutellare* independently occurred on *A. draco* with an association coefficient *V* = 0.09 (*V* ≈ 0). The dominant chigger species on *A. ilex* were *L. rusticum*, *L. densipunctatum* and *L. gongshanense*, with a total *C_r_* = 58.14% (25/43). *Leptorombidium rusticum* and *L. densipunctatum* on *A. ilex* had a slight positive association (*V* = 0.49, 0.5 < *V* < 1). All dominant chigger species were unevenly distributed among different individuals of two mouse species. Chigger infestation showed sex bias on different sexes of two mouse species. The species abundance of the chigger community on *A. draco* was revealed as a log-normal distribution pattern.

## 1. Introduction

Chigger mites belong to the families Trombiculidae and Leeuwenhoekiidae in the subclass Acari [1,2]. It is estimated that about 3013 species of chigger mites have been recorded worldwide [3,4], and over 510 species have been reported in China [3,4,5]. The life cycle of chigger mites is complex, with seven basic stages: the egg, deutovum (prelarva), larva, nymphochrysalis, nymph, imagochrysalis and adult (male and female). The larva (chiggers) is the only ectoparasitic stage in rodents and other animal hosts [1,6,7]. In addition to directly biting humans to cause chigger dermatitis, chiggers are the exclusive vector of scrub typhus (tsutsugamushi disease), which is the most important medical significance of the mites. In addition, some chigger species, e.g., *Leptotrombidium scutellare* (Nagayo, Miyagawa, Mitamura, Tamiya and Tenjin, 1921), can be potential vectors of hemorrhagic fever with renal syndrome, HFRS [4,8,9]. With the spread of scrub typhus worldwide in recent years, the prevalence of the disease in China is also on the rise [10,11].

Rodents account for more than 40% of all mammals in the world, and they are widely distributed in different environments [12,13]. In addition to being harmful to agriculture and forestry, rodents can also carry a variety of pathogens of some zoonotic diseases such as scrub typhus, leptospirosis, tularemia, HFRS and plague [3,12,13]. South China field mouse (*Apodemus draco* Barrett-Hamilton, 1900) and Lantsang field mouse (*A. ilex* Thomas, 1922) belong to the genus *Apodemus* of the family Muridae in the order Rodentia, and they are two sibling species of mice with similar morphology [14,15,16]. These two sibling rodent species were once regarded as the same species, and *A. ilex* was once considered a subspecies of *A. draco* [16,17,18]. A series of previous studies on these two mouse species have involved their taxonomic status and morphological characteristics [14,15,19,20], relative fatness and seasonal changes in digestive tract length [21,22], eco-physiology [23], karyotype [24,25] and molecular differentiation [16,26], but there is little research literature on their ectoparasites including chiggers [18,20]. To date, there have been no specific reports on chiggers of these two sibling mouse species in southwest China.

The southwest China involved in the present paper covers five provincial regions, Yunnan, Guizhou, Sichuan, Chongqing and Tibet (Xizang Autonomous Region), with a vast territory, accounting for 24.5% of China’s land area [27]. Recent studies have shown that there are obvious differences in the geographical distribution of *A. draco* and *A. ilex* in southwest China. The former is mainly distributed east of the Jinsha River (roughly equivalent to the east of 101°50′ E), while the latter is mainly distributed west of the Lantsang River (roughly equivalent to the west of 100°45′ E) [16,17,28]. Based on previous field investigations in southwest China from 2001 to 2015, this paper retrospectively studied the species composition, infestation status and some ecological characteristics (community structure and species abundance distribution) of chiggers on these two sibling species of *Apodemus* mice for the first time. As an exploration of some unknown scientific issues, this study aims to compare the difference in chigger infestation on these two sibling mouse species, enrich the knowledge of these two mouse species and their ectoparasites, and provide more scientific information for subsequent related research.

## 2. Materials and Methods

### 2.1. Field Survey Sites

The original data came from the field investigations in 91 survey sites of five provincial regions in southwest China from 2001 to 2015 [29,30]. Among the 91 sites investigated, there are 8 sites where *A. draco* was captured, and 3 sites where *A. ilex* was captured, totaling 11 sites (Figure 1).

### 2.2. Collection and Identification of Chiggers and Their Animal Hosts

Rodents and other small mammals (animal hosts) were captured with mousetraps, and chiggers on their body surfaces were routinely collected and fixed. Each host was identified into species according to its morphological appearance (body size, shape and coat color), various measurements (body length, body weight, tail length, ear height, hind foot length, etc.) and other morphological features [31,32]. The collected chiggers were mounted with Hoyer’s medium and made into slide specimens. After dehydration, drying and transparency, each chigger was identified into species under an optical microscope (Olympus Corporation, Tokyo, Japan) [1,3,33,34,35]. Based on the identification results of chiggers and their animal hosts, *A. draco* and *A. ilex*, together with their chiggers, were chosen as the target of present study. The capture and use of animals were officially approved by the local wildlife affairs authority and the Animal Ethics Committee of Dali University, and the ethics approval number is DLDXLL2020-1104.

### 2.3. Infestation Statistics Analysis

The species and numbers of all chiggers on the body surface of each *A. draco* and *A. ilex* were counted, respectively. The constituent ratio (*C_r_*), infestation prevalence (*P_M_*), mean abundance (*MA*) and mean intensity (*MI*) were adopted to calculate the infestation of mice with chiggers [7,34,36].
(1)$$C_{r}=\frac{N_{i}}{N}\times 100\%$$
(2)$$P_{M}=\frac{H_{i}}{H}\times 100\%$$
(3)$$MA=\frac{N_{i}}{H}$$
(4)$$MI=\frac{N_{i}}{H_{i}}$$

In the above formulae, *N_i_* = the number of a certain chigger species (species *i*) on a certain species of host, *N* = the total number of all the chigger species, *H* = the total number of hosts captured, *H_i_* = the number of hosts infested with chiggers.

### 2.4. Basic Community Structure Statistics

Species richness (*S*), Shannon–Wiener diversity index (*H*), Simpson dominance index (*D*) and Pielou evenness index (*E*) were used for chigger community statistics. Jaccard similarity coefficient (*J*) was used to analyze the similarity of community species composition [36,37].
(5)$$H=-\sum_{i=1}^{S}\left(\frac{N_{i}}{N}\right)\ln\left(\frac{N_{i}}{N}\right)$$
(6)$$D=1-{\sum_{i=1}^{S}\left(\frac{N_{i}}{N}\right)}^{2}$$
(7)$$E=\frac{H}{lnS}$$
(8)$$J=\frac{M}{G+F-M}$$

In the above formulae, *N* = the total number of all the chigger species, *N_i_* = the number of a certain chigger species (species *i*) on a certain species of host, *S* = species richness (the number of species), *G* = the number of chigger species in community A, *F* = the number of chigger species in community B, *M* = the number of common species existed in both community A and B. 0.00 < *J* < 0.25 means extremely dissimilar, 0.25 ≤ *J* < 0.50 means moderately dissimilar, 0.50 ≤ *J* < 0.75 means moderately similar, and 0.75 ≤ *J* < 1.00 means extremely similar.

### 2.5. Measurement of Spatial Distribution Patterns

The spatial distribution of dominant chigger species among different individuals of mouse hosts was determined by the indexes of diffusion (*C*), mean crowding (*m**) and clumping index (*I*). The calculation formulae and judgment criteria are listed in Table 1 [38,39].

### 2.6. Measurement of Interspecific Association

The association coefficient (*V*) was used to analyze the interspecific relationship between any two dominant chigger species on *A. draco* and *A. ilex*, and Chi-square test was used to test the statistical significance of *V* [5,40].
(9)$$V=\frac{ad-bc}{\sqrt{\left(a+b)(c+d)(a+c)(b+d\right)}}$$

In the above formula, *V* = association coefficient between any two chigger species, X and Y, on a certain host species; *a* = host individuals on which both chigger species X and Y concurrently appear; *b* = host individuals on which chigger species Y appears, but chigger species X does not appear; *c* = host individuals on which chigger species X appears, but chigger species Y does not appear; and *d* = host individuals on which neither chigger species X nor Y appears. When 0 < *V* ≤ 1 and *p* < 0.05, the interspecific relationship between chigger species X and Y is determined as positive association, and when −1 ≤ *V* < 0 and *p* < 0.05, negative association. When *V* = 0 or *V* ≈ 0, it can be considered that chigger species X and Y independently occur on the host.

### 2.7. Species Abundance Distribution

*X*-axis (indicating individuals of chiggers) was labeled with log intervals based on log_3_M, and *Y*-axis (representing the number of chigger species) was marked with arithmetic scales. Based on following formulae, Preston’s lognormal model was used to fit the theoretical curve of species abundance distribution of chigger community with the calculation of fitting goodness, *R*^2^ [34,41,42,43].
(10)$$S\left(R\right)=S_{0}e^{{-\left[a\left(R-R_{0}\right)\right]}^{2}}(e\;=\;2.71828\ldots )\;(\mathrm{Preston}’s\mathrm{lognormal}\;\mathrm{model})$$
(11)$$R^{2}=1-\sum_{R=0}^{m}{\left[S^{\prime }\left(R\right)-S\left(R\right)\right]}^{2}/\sum_{R=0}^{m}{\left[S^{\prime }\left(R\right)-\overset{-}{S}\left(R\right)\right]}^{2}$$
(12)$$\overset{-}{S}\left(R\right)=\frac{1}{m}\sum_{R=0}^{m}S^{\prime }(R)$$

In the above formulas, *S*(*R*) = the theoretical number of chigger species at the *R*-th log interval, *S*_0_ = the number of chigger species at the *R*_0_ log interval, *m* = the number of log intervals, *R*_0_ = the mode log interval, *S*’(*R*) = the actual chigger species at *R*-th log interval and $\overset{-}{S}\left(R\right)$ = the average chigger species for each log interval. The value of *α* was determined according to the best-fitting goodness, *R*^2^.

## 3. Results

### 3.1. Species Composition and Community Structure of Chiggers on Two Mouse Species

Among the 91 sites investigated in the five provincial regions of southwest China, there are 8 sites where *A. draco* was captured, and 3 sites where *A. ilex* was captured, totaling 11 sites. The 11 survey sites are as follows: Dali (DL), Daocheng (DC), Fuyuan (FY), Gengma (GM), Gongshan (GS), Miyi (MY), Muli (ML), Weixi (WX), Yanyuan (YY), Yongde (YD) and Zhijin (ZJ) (Figure 1).

Among the 11 survey sites shown in Figure 1, 567 *A. draco* were captured in 8 sites, and 154 *A. ilex* were captured in 3 sites, with a total of 721 hosts (567 + 154). From the body surface of 721 hosts, 313 chiggers were collected. Of the 313 chiggers collected, 290 ones were identified as 42 species and 7 genera in 3 subfamilies under 2 families, and 23 mites were unidentified because of broken bodies, dirt-covered bodies, blurred structures or suspected new species. The 23 unidentified chiggers were not included in the statistical analysis of this study. Among the 42 chigger species identified, there are 36 species on *A. draco*, 11 species on *A. ilex*, and 5 common species on both *Apodemus* (Table 2). Jaccard similarity index (*J*) shows that the species composition of chiggers on two mouse species of *Apodemus* is very different, with *J* = 0.12 (*J* < 0.25, extremely dissimilar). There are some differences in the community parameters of chiggers on two mouse species. The species richness (*S* = 36) and Shannon–Wiener diversity index (*H* = 2.89) of chigger community on *A. draco* are higher than those on *A. ilex* (*S* = 11, *H* = 2.10), but Pielou evenness index and Simpson dominance index (*E* = 0.81, *D* = 0.09) on *A. draco* are lower than those on *A. ilex* (*E* = 0.88, *D* = 0.15).

### 3.2. Overall Infestation and Dominant Species of Chiggers on Two Mouse Species

There were consistent differences in the overall infestation of chiggers in the two mouse species. Of all *A. draco* mouse hosts, 58 mice of them were infested with chiggers with 10.23% of overall infestation (*P_M_* = 10.23%, 58/567), 0.44 mites/per mouse of mean abundance (*MA* = 0.44) and 4.26 mites/per mouse of mean intensity (*MI* = 4.26), which were higher than the corresponding indices on *A. ilex* (*P_M_* = 7.14%, *MA* = 0.28 mites/mouse, *MI* = 3.91 mites/mouse). The difference in overall mean intensity (*MI*) of chiggers on two mouse species was statistically significant (*p* < 0.05), but there was no statistical significance in the differences in overall prevalence (*P_M_*) and overall mean abundance (*MA*) of the mites on the mice (*p* > 0.05). The dominant chigger species on the two mouse species were also different. The dominant chigger species on *A. draco* are *Trombiculindus yunnanus* Wang and Yu, 1965, *L. scutellare* and *L. sinicum* Yu, Yang and Gong, 1981 (total *C_r_* = 42.9%, 106/247). The dominant chigger species on *A. ilex*, however, are *L. rusticum* Yu, Yang and Gong, 1986, *L. densipunctatum* Yu, Yang and Gong, 1982 and *L. gongshanense* Yu, Yang and Gong, 1981 (total *C_r_* = 58.14%, 25/43) (Table 3). The diagnostic characteristics of the six dominant chigger species on two mouse species are listed in Table 4, and the corresponding abbreviations refer to the relevant taxonomic literature [1,2]. The photos of the representative dominant mite species are shown in Figure 2, Figure 3, Figure 4 and Figure 5.

The calculation of the Jaccard similarity index showed that the species composition of chiggers on different sexes (males and females) of two mouse species was quite different (moderately dissimilar), *J* < 0.5 (*A. draco*: *J* = 0.27, *A. ilex*: *J* = 0.45). The infestation indexes (*P_M_*, *MA*, *MI*) of chiggers were also different between the two sexes of the mice, but the differences are of no statistical significance (Table 5).

The association coefficient (*V*) between *L. sinicum* and *L. scutellare* on *A. draco* was close to 0 (*V* = 0.09, *V* ≈ 0), and that between *L. rusticum* and *L. densipunctatum* on *A. ilex* was close to 0.5 (*V* = 0.49) (Table 6 and Table 7).

All the calculated indexes of dominant chigger species on two mouse species for spatial distribution patterns were higher than the border values (*C* > 1, *m** > *m*, *I* > 1) of determining aggregated distribution (Table 1 and Table 8).

### 3.3. Species Abundance Distribution of Chigger Community

Of the 36 species and 247 identified chiggers on *A. draco*, the number of chigger individuals at Log interval 4 was the highest, but the number of species was minimal. At Log interval 0 (*R*_0_ = 0), there was only one individual chigger, but the number of chigger species was the largest (*S*_0_ = 13). The species abundance distribution of the chigger community on *A. draco* was successfully fitted by Preston’s lognormal model with *α* = 0.36 and *R^2^* = 0.86. The theoretical curve equation was (*R*) = $13e^{{-\left[0.36\left(R-0\right)\right]}^{2}}$ (*S*_0_ = 13, *R*_0_ = 0) (Table 9, Figure 6).

## 4. Discussion

Previously *A. draco* and *A. ilex* were regarded as the same species, and *A. ilex* was once considered a subspecies of *A. draco* [18,44,45]. Recent studies have proved that *A. draco* and *A. ilex* are two independent species of rodents, and they are two sibling species. Although *A. draco* and *A. ilex* are quite similar in morphology, they still have some differences in morphology, molecular characteristics and geographical distribution areas [16,17,28]. *Apodemus draco* is mainly distributed in countries and regions such as China, north Myanmar and northeast India. In China, *A. draco* is mainly distributed in Yunnan, Tibet and Fujian provincial regions. The distribution range of *A. ilex* is narrow and mainly distributed south of the Yangtze River in south China and west of the Lantsang River in southwest China [16,17,28]. The present study revealed that *A. draco* and *A. ilex* were distributed in southwest China, but they were not the dominant rodent species in this area (only 567 *A. draco* and 154 *A. ilex* were captured). The distribution range of two mouse species in southwest China was obviously different. *Apodemus draco* was mainly captured east of the Jinsha River, and *A. ilex* was mainly captured west of the Lantsang River (Figure 1). This result is consistent with some previous research reports [16,28].

The results of this study showed that *A. draco* and *A. ilex* were not only obviously different in distribution areas but also different in chigger infestations, including species composition, community structure, infestation status and dominant chigger species. The Jaccard similarity index (*J*) used in this study is an index reflecting the similarity of species composition of any two communities. When 0.00 < *J* < 0.25, the species composition of two communities is extremely dissimilar, 0.25 ≤ *J* < 0.50 means moderately dissimilar, 0.50 ≤ *J* < 0.75 means moderately similar, and 0.75 ≤ *J* < 1.00 means extremely similar [46]. The results reveal that the species composition of the chigger community on two sibling species of *Apodemus* are very different with *J* = 0.12 (*J* < 0.25), and the community parameters (*S*, *H*, *E* and *D*) are also different. The overall infestation prevalence (*P_M_*), mean abundance (*MA*) and mean intensity (*MI*) of chiggers on *A. draco* are higher than those on *A. ilex*. The dominant chigger species on *A. draco* are *T. yunnanus*, *L. scutellare* and *L. sinicum*, which are obviously different from the dominant mite species on *A. ilex* (*L. rusticum*, *L. densipunctatum* and *L. gongshanense*) (Table 3). Previous studies have shown that different small mammal species have different susceptibilities to the infestation of ectoparasites, including chiggers, which leads to differences in species composition, infestation status and dominant parasite species on different species of animal hosts [1,47]. Chevrier’s field mouse (*A. chevrieri* Miline-Edwards, 1868) is a mouse species in the same genus (*Apodemus*) as *A. draco* and *A. ilex*, and it is one of the dominant rodent species in southwest China [30]. A special study on chiggers of *A. chevrieri* in southwest China showed that its overall infestation prevalence (*P_M_* = 31.95%), mean abundance (*MA* = 6.32 mites/mouse) and mean intensity (*MI* = 19.77 mites/mouse) with chiggers were significantly higher than the corresponding infestation indexes on the two mouse species of *Apodemus* in this study. The dominant chigger species on *A. chevrieri* are *L. scutellare*, *L. densipunctatum* and *L. cricethrionis* Wen, Sun and Sun, 1984, which are also obviously different from *A. draco* and *A. ilex* in this study [30]. The above differences reflect the different susceptibility of different mouse host species to chigger infestation. In this study, there are a series of differences in species composition, infestation status and dominant species composition of chiggers between two mouse species of *Apodemus*, which further verify the different susceptibility of different hosts to chiggers and the different preference of chiggers to different hosts. From the aspect of ectoparasites, the differences in species composition, infestation status and dominant species composition of chiggers between two sibling mouse species also support that *A. draco* and *A. ilex* belong to two independent species [16,28]. As one of the dominant chigger species of *A. draco*, *L. scutellare* is not only the second major vector of scrub typhus in China but also a potential vector of hemorrhagic fever with renal syndrome [9,48]. The occurrence of *L. scutellare* on the body surface of *A. draco* may increase the potential risk of spreading the pathogen of scrub typhus, *Orientia tsutsugamushi*, from rodents to humans.

The calculation results of the Jaccard similarity index (*J*) showed that the species composition of chiggers on different sexes of two mouse species, *A. draco* and *A. ilex*, were quite different (*J* < 0.5, moderately dissimilar). Besides, the infestation indexes (*P_M_*, *MA*, *MI*) of chigger on different sexes of two mouse species were also different (Table 5). These results indicate that there is sex bias in chigger infestation between male and female hosts [29,36]. Sex bias is prevalent in parasite infections (including ectoparasite infestations), and many parasites are more likely to choose to parasitize in male hosts [4,49,50], but the preference of chiggers to male hosts is not obvious in this paper (Table 5), which may need further studies.

In this study, the interspecific relationship between the dominant chigger species on two mouse species was measured by association coefficient (*V*). The results showed that the association coefficient (*V*) between *L. sinicum* and *L. scutellare* on *A. draco* was close to 0 (*V* = 0.09, *V* ≈ 0). The *V* between *L. rusticum* and *L. densipunctatum* on *A. ilex* was close to 0.5 (*V* = 0.49, *V* ≈ 0.5). The association coefficient (*V*) used in this paper is one of the simple and practicable methods to judge the interspecific relationship between any two species in a certain community. The range of *V* is from 0 to positive and negative 1, that is, [1, ±1]. When *V* approaches 0, the distribution of the two species is independent of each other. When *V* is positive and close to 1, it means that two species have a tendency to coexist in a certain environment or on a certain species of host for parasites. When the *V* is negative and close to -1, it means that there is a mutually exclusive relationship between the two species [5,40]. This study implies that the distribution of *L. sinicum* and *L. scutellare* on *A. draco* seems independent of each other without obvious interspecific dependency. A low degree of interdependence, however, exists between *L. rusticum* and *L. densipunctatum* on *A. ilex*, and these two chigger species tend to choose the same individuals of *A. ilex* at the same time, but the degree of interdependence is still relatively low (*V* < 0.5).

The results showed that all the distribution indexes calculated were larger than the critical value (*C* > 1, *m** > *m*, *I* > 1) of determining the aggregated distribution (Table 1, Table 8), and this indicates that the dominant chigger species are of aggregated distribution among different individuals of their corresponding mouse host, *A. draco* and *A. ilex*. This aggregated distribution further indicates that the distribution of dominant chigger species among different individuals of their hosts is very uneven. Some host individuals may harbor a large number of chiggers on their body surface, while some other hosts may have no or only a few chiggers. The aggregated distribution of ectoparasites, including chiggers, suggests that there is an intraspecific relationship of mutual attraction and interdependence between different individuals of the same parasite species. This mutual attraction and interdependence within a certain species are conducive to the survival, mating and reproduction of the population [5,51].

The species abundance distribution aims to reveal the relationship between the number of species and the number of individuals in a community, which reflects the proportion structure of common and rare species in the community [33,42,43]. In the present study, the species abundance distribution of the chigger community on *A. draco* was successfully fitted by Preston’s lognormal model, which shows that most chigger species are rare species with few individuals, while few mite species are dominant species with abundant individuals. With the increase of chigger individuals, the number of chigger species gradually decreased (Table 9, Figure 6). The result is consistent with some previous reports on the species abundance of chiggers [36,43]. Due to the small number of individuals and species of chiggers and mice collected, however, Preston’s lognormal model is not applicable to the chigger community on *A. ilex*. In ecological practice, if the species abundance distribution of a specific community is successfully fitted by Preston’s log-normal distribution model, the number of expected total species in the community can be roughly estimated, but this estimation must be based on a large number of samples [52]. Due to the small number of host samples (only 567 *A. draco* and 154 *A. ilex* captured) and a small number of chiggers (total 290), the total number of chigger species was not estimated in this paper, which remains to be conducted in further studies.

It must be pointed out that the present study is just a preliminary comparison of chiggers on two sibling mouse species due to the small number of host samples, especially *A. ilex*, and some results may still be unstable. With the expansion of survey areas and the increase of host samples in future research, some results may fluctuate and change to some extent.

## 5. Conclusions

In southwest China, the susceptibility of two sibling mouse species (*A. draco* and *A. ilex*) to chigger infestation is quite different, with different species composition and community structure, different infestation status and different dominant chigger species. The results support that *A. draco* and *A. ilex* are two independent rodent species from the aspect of chigger infestation.

## Figures and Tables

**Figure 1 animals-13-01480-f001:**
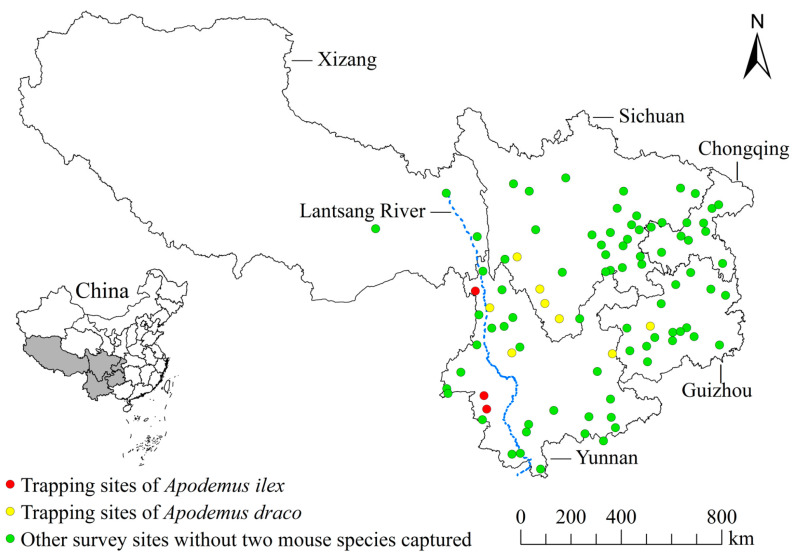
Trapped sites of *Apodemus draco* and *A. ilex* in southwest China (n = 11).

**Figure 2 animals-13-01480-f002:**
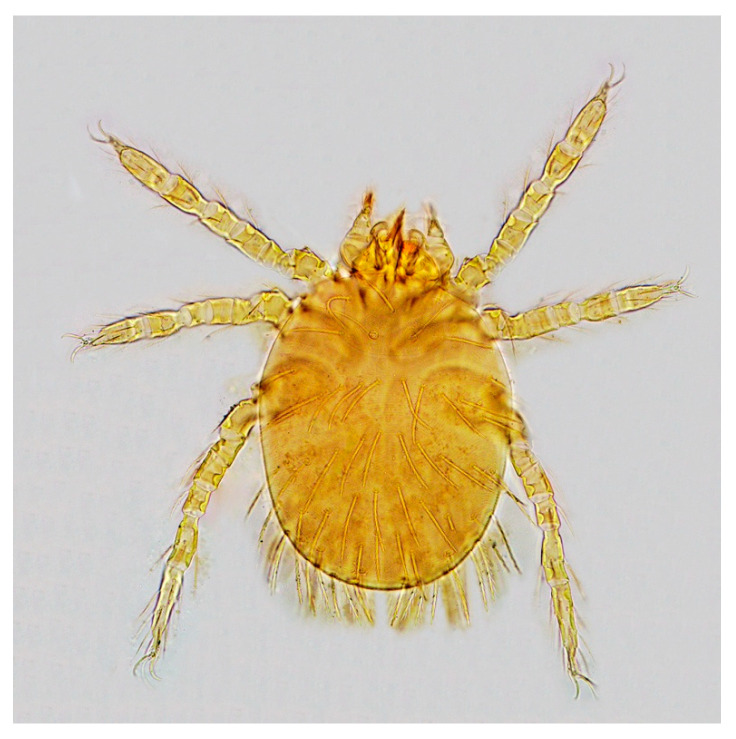
*Leptotrombidium scutellare* (Nagayo et al., 1921) [1,2] (10 × 40).

**Figure 3 animals-13-01480-f003:**
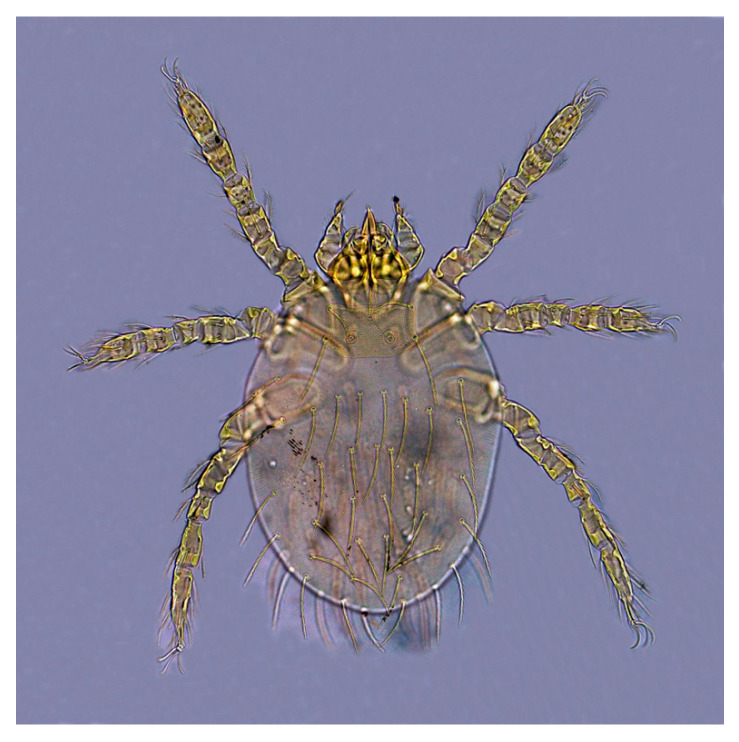
*Leptotrombidium densipunctatum*. Yu et al., 1982 [1,2] (10 × 40).

**Figure 4 animals-13-01480-f004:**
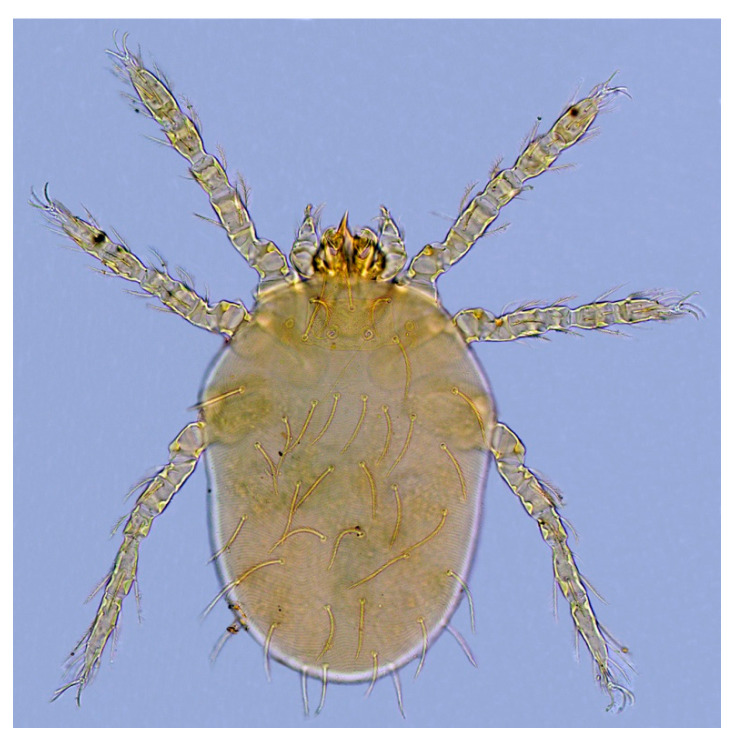
*Leptotrombidium sinicum* Yu et al., 1981 [1,2] (10 × 40).

**Figure 5 animals-13-01480-f005:**
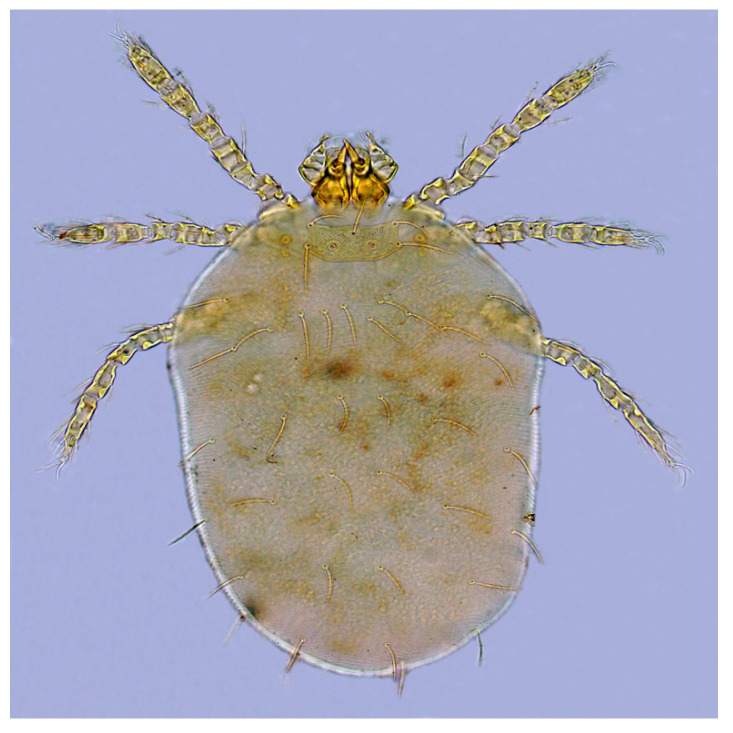
*Leptotrombidium rusticum* Yu et al., 1986 [1,2] (10 × 40).

**Figure 6 animals-13-01480-f006:**
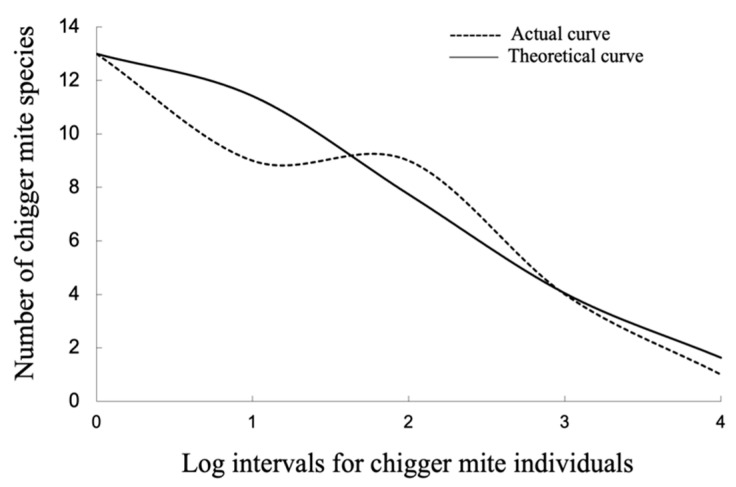
Theoretical curve fitting of species abundance distribution of chigger community on Apodemus draco in southwest China (2001–2015).

**Table 1 animals-13-01480-t001:** Index formulae and judgment criteria for spatial distribution patterns.

Name	Formula	Aggregation Distribution	Random Distribution	Uniform Distribution
*C*	$C=\sigma^{2}/m$	>1	=1	<1
*m**	$m^{*}=m+\sigma^{2}/m-1$	*>m*	*=m*	<*m*
*I*	$I=n\times \frac{\left(\sum m^{2}-\sum m\right)}{\left[{\left(\sum m\right)}^{2}-\sum m\right]}$	>1	=1	<1

$Annotation:\sigma^{2}=$ variance, *m* = mean and *n* = the total number of host samples.

**Table 2 animals-13-01480-t002:** Distribution of identified 42 chigger species on two mouse species (*Apodemus draco* and *A. ilex*) at 11 survey sites of southwest China (2001–2015).

Chigger Names	Distribution on Hosts		Distribution at Survey Sites	Chigger Names	Distribution on Hosts		Distribution at Survey Sites
	*A. draco*	*A. ilex*			*A. draco*	*A. ilex*	
**Family Trombiculidae** [1,2]				*Trombiculindus yunnanus* Wang et Yu, 1965 [1]	+	+	YY, GS
**Subfamily Trombiculinae** [1,2]				*T. cuneatus* Traub et Evans, 1951 [1]	-	+	GS
*Leptotrombidium scutellare* (Nagayo et al., 1921) [1,2]	+	+	GS, WX	*T. nujiange* Wen et Xiang, 1984 [1]	-	+	GS
*L. sinicum* Yu et al., 1981 [1,2]	+	-	WX	*T. bambusoides* Wang et Yu, 1965 [1]	+	+	ZJ, MY, YY, GS
*L. rusticum* Yu et al., 1986 [1,2]	-	+	GS	*Neotrombicula japonica* (Tanaka et al., 1930) [1]	+	-	MY
*L. wangi* Yu et al., 1986 [1,2]	+	-	WX	*N. tongtianhensis* Yang et al., 1995 [1]	+	-	WX
*L. densipunctatum* Yu et al., 1982 [1,2]	+	+	GS, WX	*N. aeretes* Hsu et Yang, 1985 [1]	+	-	WX
*L. robustisetum* Yu et al., 1983 [1,2]	+	-	WX	*Helenicula simena* (Hsu et Chen, 1957) [1]	+	-	MY, YY, ML
*L. laojunshanense* Yu et al., 1986 [1,2]	+	-	WX	*H. hsui* Zhao, 1990 [1]	+	-	YY
*L. baoshui* Wen et Xiang, 1984 [1,2]	+	-	ML	**Subfamily Gahrliepiinae** [1]			
*L. kawamurai* (Fukuzumi et Obata, 1953) [1,2]	+	-	YY				
*L. alpinum* Yu et Yang, 1986 [1,2]	+	-	YY, WX	*Walchia kor* (Chen et Hsu, 1957) [1]	+	-	ML
*L. gongshanense* Yu et al., 1981 [1,2]	+	+	YY, GS	*W. zangnanica* Wu et Wen, 1984 [1]	+	-	MY
*L. muntiaci* Wen et Xiang, 1984 [1,2]	+	-	ML	*W. tianguangshanensis* Zhao et al., 1980 [1]	-	+	GM
*L. jinmai* Wen et Xiang, 1984 [1,2]	+	-	YY	*W. ewingi* (Fuller, 1949) [1]	-	+	GS
*L*. *eothenomydis* Yu et Yang, 1986 [1,2]	+	-	ZJ, YY, WX	*Gahrliepia myriosetosa* Wang, 1964 [1]	+	-	ML
*L. nyctali* Wen et Sun, 1984 [1,2]	+	-	YY	*G. linguipelta* Jeu et al., 1983 [1]	+	-	YY, WX
*L. suense* Wen, 1984 [1,2]	+	-	ML	*G. longipedalis* Yu et Yang, 1986 [1]	+	-	WX
*L. neotebraci* Xiang et Wen, 1986 [1,2]	+	-	ML	*G. radiopunctata* Hsu et al., 1965 [1]	-	+	GM
*L. rupestre* Traub et Nadchatram, 1967 [1,2]	+	-	ML	*G. megascuta* Hsu et al., 1965 [1]	+	-	WX
*L. yongshengense* Yu et Yang, 1986 [1,2]	+	-	MY, ML	**Family Leeuwenhoekiidae** [1]			
*L. longimedium* Wen et Xiang, 1984 [1,2]	+	-	ML	**Subfamily Leeuwenhoekiinae** [1]			
*L. sinotupaium* Wen et Xiang, 1984 [1,2]	+	-	ML	*Chatia alpine* Shao et Wen, 1984 [1]	+	-	YY
*L. bambicola* Wen et Xiang, 1984 [1,2]	+	-	ML	*C. acrichela* Wen et al., 1984 [1]	+	-	YY

Annotation: “+”= The host surface contains this chigger mite. “-”= The host body surface does not contain this chigger mite.

**Table 3 animals-13-01480-t003:** Dominant chigger species on two species of mice (*Apodemus draco* and *A. ilex*) in southwest of China (2001–2015).

Names of Mouse Hosts	Names of Dominant Chigger Species	Individuals and Constituent Ratios (*C_r_*) of Chiggers		Overall Infestations of Chiggers on the Hosts		
		Individuals	*C_r_* (%)	*P_M_* (%)	*MA*	*MI*
*A. draco* (n = 567)	*T. yunnanus*	50	20.24	0.35	0.09	25.00
	*L. scutellare*	34	13.77	3.02	0.06	2.00
	*L. sinicum*	22	8.91	0.89	0.04	4.40
*A. ilex* (n = 154)	*L. rusticum*	12	27.91	2.60	0.08	3.00
	*L. densipunctatum*	8	18.60	2.60	0.05	2.00
	*L. gongshanense*	5	11.63	1.30	0.03	2.50

**Table 4 animals-13-01480-t004:** The diagnostic characteristics of the six dominant chigger species on two mouse species, *Apodemus draco* and *A. ilex*.

Names of Mouse Hosts	Names of Dominant Chigger Species	Diagnostic Characteristics
*A. draco*	*T. yunnanus*	fPp = N/N/BNN; PC = 3; Gn = 2; Sc: PL > AM> AL; PL/SB; fCx = 1.1.1; fSt = 2.2; fD = 2H + 8-6-6-6-2-2; DS = 32; VS = 35; NDV = 67; Ip = 1059; AW 88, PW 99, SB 41, ASB 32–33, PSB 23, SD 55, AP 23, AM 55, AL 42, PL 65 S 75, H-, D_min_-, D_max_-, V_min_-, V_max_-, pa 353, pm 323, pp 383.
	*L. scutellare*	fPp = N/N/BNN; PC = 3; Gn = 2; Sc: PL > AM> AL; SB-PL; fCx = 1.1.1; fSt = 2.2; fD = 2H-10-[10-2]-[12-4]-8-6-2; DS = 56; VS = 38; NDV = 94; Ip = 858; AW 71–72, PW 79–82, SB 30–32, ASB 29–32, PSB 15–17, SD 44 49, AP 28–29, AM 57–59, AL 50–51, PL 62–64, S 78–86, H 62, D_min_ 44, D_max_ 57–69, V_min_ 34, V_max_ 50, pa 295, pm 256, pp 307.
	*L. sinicum*	fPp = N/N/BNN; Pc = 3; Gn = 2; fSc: PL > AM > AL; PL/SB; fCx = 1.1.1; fSt = 2.2; fD = 2H-9-8-8-8-2-3; DS = 38 41; VS = 36–39; NDV = 78; Ip = 824–893; AW 60–67, PW 73–76, SB 25–30, ASB 35–37, PSB 14–15, SD 49–52, AP 30–34, AM 60–67, AL 51–60, PL 70–77, S 73–79, H 68–74, D_min_ 50–55, D_max_ 68–72, V_min_ 35–42, V_max_ 45–56, pa-, pm-, pp-.
*A. ilex*	*L. rusticum*	fPp = N/N/BNN; Pc = 3; Gn = 2; fSc: PL > AM >> AL; PL/SB; fCx = 1.1.1; St = 2.2; D = 2H-8-6-6-4-2; DS = 30; VS = 26–30; NDV = 56; Ip = 685–725; AW 64–70, PW 71–80, SB 31–37, ASB 24–28, PSB 13–16, SD 40, AP 18–22, AM 41 = 55, AL 30–38, PL 53–65, S 55–70, H 48–58, D_min_ 38–48, D_max_ 51–61, V_min_ 20–30, V_max_ 43–54, pa 236, pm 217, pp 246.
	*L. densipunctatum*	fPp = N/N/BNN; Pc = 3; Gn = 2; fSc: PL > AM > AL; SB-PL; fCx = 1.1.1; (St = 2.2; fD = 2H1-8-6-6-4-2-2; DS = 28–30; VS = 27–35; NDV = 60; Ip = 776; AW 68–79, PW 80–88, SB 33–38, ASB 29–33, PSB 13–15, SD 41–48, AP 25–31, AM 58–71, AL 43–50, PL 65–74, S 80–87, H 60–73, D_min_ 45–55, D_max_ 50–65, V_min_ 31–35, V_max_ 45–57, pa 243–293, pm 220–293, pp 257–304.
	*L. gongshanense*	fPp = N/N/BNN; Pc = 3; Gn = 2; fSc: AM > PL > AL; PL/SB; fCx = 1.1.1; fSt = 2.2; fD = 2H-8-6-6-4-2; DS = 28–30; VS = 22–25; NDV = 53; Ip = 675–710; AW 65–70, PW 73–78, SB 30–33, ASB 25–27, PSB 12–14, SD 37–40, AP 22–23, AM 64–76, AL 38–42, PL 60–66, S 78–84, H 54–61, D_min_ 42–47, D_max_ 52–57, V_min_ 37–40, V_max_ 42–47, pa-, pm-, pp-.

**Table 5 animals-13-01480-t005:** Infestation differences of chiggers on two sexes of hosts, *Apodemus draco* and *A. ilex*.

Host Names	Host Sexes	Number of Hosts	Species, Individuals and Constituent Ratios (*C_r_*) of Chiggers			Overall Infestations of Chiggers on the Hosts		
			Species	Individuals	*C_r_* (%)	*P_M_* (%)	*MA*	*MI*
*A. draco*	Female	184	17	57	23.85	10.33	0.31	3.00
	Male	378	25	182	76.15	9.79	0.48	4.92
	Total	562	-	239	100.00	20.12	0.79	7.92
*A. ilex*	Female	86	10	32	74.42	8.14	0.37	4.57
	Male	68	6	11	25.58	5.88	0.16	2.75
	Total	154	-	43	100.00	14.02	0.53	7.32

Annotation: There were five individuals (n = 5) of *A. draco* without sex records, and these five mice were not included in the calculation of the above table.

**Table 6 animals-13-01480-t006:** Interspecific association coefficient (*V*) between two dominant chigger species, *Leptotrombidium sinicum* and *L. scutellare*, on *Apodemus draco* in southwest China (2001–2015).

Dominant Chigger Species		*L. sinicum* (Species X)		
		+	-	Total
*L. scutellare* (species Y)	+	1 (*a*)	16 (*b*)	17 (*a + b*)
	-	4 (*c*)	548 (*d*)	552 (*c + d*)
Total		5 (*a + c*)	564 (*b + d*)	569 (*n*)
Association coefficient		*V* = 0.09		
Chi-square		*χ*^2^ = 5.037		
Significance		*p* < 0.05		

Annotation: “+” = The host surface contains this chigger mite. “-” = The host body surface does not contain this chigger mite.

**Table 7 animals-13-01480-t007:** Interspecific association coefficient (*V*) between two dominant chigger species, *Leptotrombidium rusticum* and *L. densipunctatum*, on *Apodemus ilex* in southwest China (2001–2015).

Dominant Chigger Species		*L. rusticum* (Species X)		
		+	-	Total
*L. densipunctatum* (species Y)	+	2 (*a*)	2 (*b*)	4 (*a + b*)
	-	2(*c*)	152 (*d*)	154 (*c + d*)
Total		4(*a + c*)	154 (*b + d*)	158 (*n*)
Association coefficient		*V* = 0.49		
Chi-square		*χ*^2^ = 37.475		
Significance		*p* < 0.05		

Annotation: “+” = The host surface contains this chigger mite. “-” = The host body surface does not contain this chigger mite.

**Table 8 animals-13-01480-t008:** Spatial distribution indexes of dominant chigger species on two mouse species (*Apodemus draco* and *A. ilex*) in southwest China (2001–2015).

Names of Mouse Hosts	Names of Dominant Chigger Species	*C*	*I*	*m**
*A. draco*	*T. yunnanus*	26.40	25.40	25.49
	*L. scutellare*	4.18	3.18	3.24
	*L. sinicum*	14.90	13.90	13.94
*A. ilex*	*L. rusticum*	4.95	3.95	4.03
	*L. densipunctatum*	2.71	1.72	1.77
	*L. gongshanense*	3.38	2.39	2.42

**Table 9 animals-13-01480-t009:** Fitting results of species abundance distribution of chigger community on *Apodemus draco* in southwest China (2001–2015).

Log Intervals Based on log_3_M	Individual Ranges of Chiggers in Each Log Interval	Midpoint Values of Each Individual Range	*A. draco*	
			Actual Chigger Species	Theoretical Chigger Species
0	0–1	1	13	13
1	2–4	3	9	11.42
2	5–13	9	9	7.74
3	14–40	27	4	4.04
4	41–121	81	1	1.63

## Data Availability

The experimental data used to support the findings of this study are available from the corresponding author upon request.

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
