# Peer review of "Comparison of Chiggers (Acari: Trombiculidae, Leeuwenhoekiidae) on Two Sibling Mouse Species, *Apodemus draco* and *A. ilex* (Rodentia: Muridae), in Southwest China"

_animals, 2023, doi:10.3390/ani13091480_

Round 1

Reviewer 1 Report

The ms is valuable, and good in the shape. I am positive and only made minor comments/revisions. Congratulations!

Reviewer 2 Report

Overall, the paper provides a comparative analysis of chigger infestations on two distinct species of rodents in southwest China, offering insights and information for future research on chiggers and their infestations on rodents. However, given the study period of 2001-2015, two concerns must be addressed. Firstly, it is essential to clarify whether any field investigations have been carried out after 2015, and if so, why the updated data has not been included in the manuscript. Secondly, it is necessary to specify whether any part of the data has been previously published, and if so, the differences between this paper and the previous publication must be explicitly stated.

 Furthermore, there are some key areas that can be improved to enhance the quality of the paper. Firstly, the study could explore the variations in chiggers and rodents across different habitats (e.g., residential areas, cultivated farmlands, grasslands, shrubs, and forests), providing an opportunity for comparative analysis. Secondly, the manuscript could benefit from investigating the reasons behind the differences in chigger species composition and the parameters of the chigger community between the two rodent species. These potential areas of research can significantly enhance the paper's quality and make a significant contribution to the field.

Reviewer 3 Report

Dear Editor and authors.

This exciting manuscript compares the parasite load of chiggers in two rodents from China. The manuscript needs to be improved in the taxonomic part. As a taxonomist, I strongly recommend that authors review species names (the correct spelling), include the author and year of the species when cited for the first time, and not use the first abbreviated epithet when mentioned for the first time. Furthermore, I would like the authors to include the diagnosis for each species of chigger found, especially for complex genera (e.g., Leptotrombidium), and provide diagnostic photos of the mounted slides.

Reviewer 4 Report

The paper is written correctly, however, some mistakes are overlooked. I add all of my comments to the body of the manuscript.

Round 2

Reviewer 3 Report

Dear editor. I considered the manuscript accepted for publication.

Author Response

Many thanks to the reviewer for the comments and suggestions.